# Information Disclosure Impacts Intention to Consume Man-Made Meat: Evidence from Urban Residents’ Intention to Man-Made Meat in China

**DOI:** 10.3390/ijerph20042950

**Published:** 2023-02-08

**Authors:** Yong Chen, Li Liu, Chao Li, Yangfei Huang, Qiaojie Luo

**Affiliations:** 1School of Civil Engineering & Architecture, Zhejiang University of Science & Technology, Hangzhou 310023, China; 2School of Economics and Social Welfare, Zhejiang Shuren University, Hangzhou 310015, China; 3Zhijiang College, Zhejiang University of Technology, Shaoxing 312030, China; 4Stomatology Hospital, School of Stomatology, Zhejiang University School of Medicine, Hangzhou 310006, China

**Keywords:** healthy economy, man-made meat, information disclosure, awareness–behavior relationship, consume intention, moderating effect

## Abstract

Meat substitutes such as man-made meat are emerging to promote low-carbon healthy consumption, mitigate climate change, and assist healthy economic development; however, most consumers seem reluctant to make the transition. While profound social change may be required to make significant progress in this area, limited efforts have been made to understand the psychological processes that may hinder or facilitate this transition. To clearly identify the factors influencing the public’s intention to consume man-made meat and their influencing paths, this study analyzes the mechanism by which man-made meat information disclosure affects the public’s intention to consume these products based on the social cognitive theory of “awareness-situation-behavior” and using structural equation modeling, with residents of seven Chinese cities as examples (647 respondents). The results of this study yielded three main findings. First, low-carbon awareness, personal social responsibility awareness, and man-made meat risk perception significantly influence the public’s intention to consume man-made meat, with risk perception having the greatest influence (−0.434). Second, low-carbon awareness and man-made meat risk perception have a significant interaction effect on the public’s intention to consume man-made meat (−0.694). Third, man-made meat information disclosure has the most significant moderating effect on the relationship between low-carbon awareness and the public’s intention to consume man-made meat, as well as a moderating effect on the relationship between man-made meat risk perception and the public’s intention to consume man-made meat.

## 1. Introduction

Global climate change is the largest, widest, and most far-reaching challenge facing mankind to date, and one of the most important factors influencing the healthy economic and social development of the world, as well as reshaping the global political and economic landscape. The international community has come to a consensus about the need to curb global warming and reduce carbon emissions in the 21st century [1]. China, the largest developing country, faces dual challenges of sustainable economic development and climate change response. To actively respond to climate change, China has successfully signed the United Nations Framework Convention on Climate Change, the Kyoto Protocol, and the Paris Agreement and has fully, effectively, and continuously implemented the highest political commitments to all aspects of these responses [2]. Given that fossil fuels—including various types of fuel oils, gas, paraffin, coal, and natural gas—are the main source of carbon emissions, it is crucial to implement measures such as optimizing the industrial structure, adjusting the energy consumption structure, strengthening the industrial application of new-generation information technology, and improving energy resource utilization efficiency, in order to reduce carbon emissions [3]. However, as food is the basis for human survival, the carbon emissions generated by dietary consumption are increasingly significant. According to a recent report by EAT, a non-profit organization in Oslo, Norway, carbon dioxide (CO_2_) emissions from human society’s dietary patterns account for about 24% of global CO_2_ emissions, equivalent to the total CO_2_ emissions from electricity production activities [4].

In recent years, the quality of life among residents of China has continuously improved, and there has been a growing trend of consuming meat instead of grain. However, the greenhouse gas emissions factor of animal-based meat is significantly higher than those of man-made meat, as animal husbandry not only produces a large number of greenhouse gases but also depletes forests and grasslands that absorb such gases [5]. It is thus important to urgently address the problem of effective reduction of food-related carbon emissions.

In the context of the “dual carbon” (carbon peaking and carbon neutrality) goal, an increasing number of studies have shown that man-made meat has great potential to reduce food’s carbon footprint [6]. For example, man-made meat involves no farming process, no physiological carbon emissions, and a high energy conversion rate. According to the China Man-made Meat Carbon Reduction Insight Report, man-made meat can save more than 90% of carbon emissions compared with similar animal meat products [7]. Man-made meat also has the advantages of vegetable food with the taste and texture of animal meat, which is in line with the trend to upgrade consumption. In practice, however, the promotion of man-made meat in China is very difficult compared to that in other countries. Studies have shown that the lower acceptance of man-made meat is the biggest obstacle to its promotion [8]. As direct consumers of man-made meat, the public often accepts it less than animal meat ingredients, considering factors such as nutrition and the health risks of man-made meat. Scientifically and effectively improving residents’ intention to accept man-made meat is an effective way in its promotion.

Previous research on the influencing factors of the public’s intention to consume man-made meat has revealed that there are many such factors, focusing on the ingredients, processing options, taste, and price of man-made meat [9]. Some studies have also revealed that the public is reluctant to consume man-made meat because of concerns that it is unnatural and unhealthy. Other studies have noted the influence of sociodemographic indicators such as gender, income, occupation, and dietary habits on residents’ intention to consume man-made meat [10]. The influence of public psychological perception on their intention to consume man-made meat is also a hot topic of interest for scholars [11].

A literature review revealed that most studies have focused on determining the factors influencing the public’s intention to consume man-made meat, whereas few have explored the relationship between these factors from an integrated perspective and the pathways through which they act on consumption intention. Moreover, public awareness or quality remains a weak link in the current environment of climate change and resource utilization and has become one of the biggest barriers to low-carbon development in China. Raising public awareness of low-carbon living and promoting public low-carbon consumption behavior has become a crucial strategic task and central mission. So far, few studies have been conducted that consider the influence of psychological factors on low-carbon consumption intention from the perspective of public awareness. Behavioral decisions (or behavioral intentions) are governed by the public’s perception of things, which in turn is influenced by the external environment. Therefore, this study takes information disclosure (ID) as an external variable. With the help of the “awareness-situation-behavior” theory [12], it is of great value to comprehensively study the influence of ID on the public’s intention to consume man-made meat.

According to the awareness-situation-behavior theory, “awareness” has a direct influence on behavior. At the same time, the relationship between “awareness–behavior” is moderated by internal and external “situation” factors. This provides a theoretical basis for studying the impact of ID on the public’s intention to consume man-made meat. As direct consumers of man-made meat, the public behaves under the influence of their own low-carbon awareness, social responsibility awareness, and attitudes toward new man-made meat ingredients. The public’s behavioral intention to consume man-made meat needs to be explored in terms of their perceptions of climate change and energy issues, as well as the influences of low-carbon awareness, social responsibility awareness, and awareness of the risks associated with man-made meat on their behavioral decisions.

The symmetry and amount of information disclosed can significantly affect the public’s perception of the environment, as people do not always make rational judgments about such matters. The extent to which the public has relevant information plays a large role in their awareness and even directly influences their decisions about things [13]. Current research on the influence of ID on public or group behavior has primarily focused on resources and the environment [14,15], while little research has been conducted on the influence of ID on low-carbon consumer products on public behavior. Therefore, this paper chooses a low-carbon consumer product, man-made meat, as an example for analysis. To comprehensively analyze the factors that affect the public’s intention to consume man-made meat and the associated mechanisms, the degree of man-made meat ID is introduced into the study. This study is conducive to the formulation of low-carbon food consumption policies and the further improvement of residents’ low-carbon awareness to promote the realization of the “dual carbon” goal.

On this basis, this study explores how the degree of ID about man-made meat affects residents’ intention to consume man-made meat and proposes countermeasures to promote residents’ low-carbon food consumption. This is a supplement to previous research and can better provide policy recommendations for low-carbon food consumption habits, as well as guide management practices.

## 2. Literature Review and Hypotheses

### 2.1. Literature Review

To explore the influence of the public’s low-carbon awareness and perception of man-made meat on the behavioral intention to consume man-made meat, the psychological factors and their ways of operation underlying the public’s behavioral intention should first be fully understood. Early psychologists experimentally studied the influence of individual intrinsic factors (e.g., perception, cognition, emotion, motivation, and attitude, as well as gender and age) and external environment (e.g., institutional structure, technological level, economic status, and cultural background) on the mode and path of individual behaviors. After summarizing and sorting out various influencing factors, the classic Lewin behavior model concluded that individual behavior is the result of the interaction between individuals and their situation [16].

The theory of rational behavior and the theory of planned behavior were then successively put forward. The theory of rational behavior holds that individual behavior depends on specific behavioral intentions, which are the result of a combination of personal attitudes and subjective norms [17]. The formation of attitudes can be explained on two levels: the individual’s salient beliefs about the outcome of performing a particular behavior and the evaluation of the outcome. Subjective norms or social attitudes are an individual’s perception of the pressure from important people or organizations around him or her to perform or not perform a particular behavior. The theory of rational behavior states that any factor can only indirectly influence behavior through attitudes and subjective norms. The theory of planned behavior builds on the theory of rational behavior by introducing perceived behavioral control, which is the degree of control (or mastery) that an individual expects to feel when engaging in a particular behavior and is similar to the concept of self-efficacy or enabling conditions [18]. However, regarding the influence of individual attitudes, the theory of rational behavior and the theory of planned behavior only emphasize the instrumental components of attitudes (e.g., useful–harmful and valuable–worthless) but ignore the emotional components of attitudes, which to some extent limits the ability of the theories to explain behavior.

Regarding the relationship between individual attitudes and environmental behavior, the attitude-situation-behavior theory argues that environmental behavior is the product of the interaction between environmental attitudes and situational factors [13]. An empirical study on curbside recycling by Guagnano et al. [19] demonstrated that when the influence of situational factors is neutral, the relationship between environmental attitudes and environmental behavior is the strongest; when situational factors are advantageous or disadvantageous, they may greatly promote or prevent the environmental behavior, and in the studied case the influence of environmental attitudes on environmental behavior was close to zero. Although the attitude-situation-behavior theory has been applied to determine the influence of individual intrinsic attitude factors and external situational factors on behavior, as well as verifying the moderating effect of situational factors on environmental attitudes and environmental behavior, no deeper analysis of the attitude formation process and the mechanism by which attitudes influence behavior has ensued. The awareness-situation-behavior model proposes that low-carbon psychological awareness is the antecedent variable of low-carbon consumption behavior, while the awareness–behavior relationship is moderated by situational factors. However, the model is still under exploration and needs to be verified with large sample data. This study investigates the mechanism by which ID influences the intention to consume man-made meat based on the awareness-situation-behavior theoretical model, which not only clarifies how the public’s low-carbon food consumption behavior is influenced but also has considerable implications for deepening the research on how such mechanisms influence individual behavior.

### 2.2. Hypotheses

Within the context of the literature review, this study hypothesizes that climate change and energy issue perception, low-carbon awareness, individual social responsibility awareness, and perception of the risks of man-made meat have direct influences on the public’s intention to consume man-made meat and that the awareness–behavioral–intention relationship is moderated by an external situational variable (ID related to man-made meat). The path of awareness (climate change and energy issue perception, low-carbon awareness, individual social responsibility awareness, and man-made meat risk perception)—situation (man-made meat information disclosure)—behavioral intention (man-made meat consumption intention) was constructed to analyze the mechanism by which ID related to man-made meat influences the public’s intention to consume such products.

#### 2.2.1. Direct Influence of Awareness on Behavioral Intention

According to previous studies, awareness factors such as emotion, responsibility, and risk may have significant influences on behavior. Emotional factors mainly refer to the public’s perception, guilt, or sensitivity to climate change and environmental issues. Dispoto [20] conducted a correlation analysis of environmental emotion and environmental behavior, showing the presence of a strong positive correlation between the two. Bamberg and Möser [21] carried out a meta-analysis of multiple studies on the determinants of pro-environmental behavior and found that individual guilt (as a type of emotion) is a significant predictor of pro-environmental behavior. Based on a study of the emotional characteristics of consumer recycling behavior, Meneses [22] found that this behavior is more related to positive emotions than negative ones. In their study of factors influencing urban residents’ environmental behavior, Sun et al. [23] found that Chinese residents are more emotionally concerned about ecological issues than residents in Western countries. This positive emotional factor is a good foundation based on which positive environmental behavior can be promoted with proper education and publicity. Based on a study of survey data, Huang [24] reported that travelers’ pro-environmental awareness increases their low-carbon consumption behavior. Koenig-Lewis et al. [25] found that emotion effectively moderates environmental concerns and cognitive benefits, as well as influences consumers’ behavioral intentions to purchase pro-environmental packaged products, which in turn drives consumers’ low-carbon consumption behavior. Ouyang et al. [26] analyzed the environmental awareness, environmental protection behavior measures, and influencing factors of Chinese urban residents and found that awareness plays a positive role in environmental protection behavior. In studying the influence of resource conservation awareness on such behavior, Wang [12] constructed a theoretical, conceptual model of awareness-behavior and verified that awareness plays a great role in influencing resource conservation behavior. Pan and Wang [27] investigated the factors influencing tourists’ environmental and behavioral intentions, concluding that such intentions are directly influenced by the behavioral control of specific environmental perceptions. Most of the existing studies support that environmental emotion influences environmental behavior; that is, the public’s perception of environmental issues and environmental protection awareness have significant impacts on the intention of environmental behavior.

Regarding responsibility awareness, Wells et al. [28] quantitatively studied the relationship between consumer behavior and responsibility awareness, and the results showed that there is indeed a relationship between them. Bouman et al. [29] argued that individuals with a high level of personal responsibility are more inclined to take action when climate change concerns lead to climate action. Jakučionytė-Skodienė and Liobikienė [30] studied the climate change mitigation concerns, personal responsibility, and actions of residents of EU countries and found that taking personal responsibility has a significant and positive influence on almost all individual actions related to climate change mitigation. Lu and Sun [31] argued that the public’s concern for the environment and pro-environmental responsibility are the key factors influencing—and are directly proportional to—their pro-environmental behavior. Yu [32] studied the factors influencing the consumption behavior of pro-environmental products, showing that environmental responsibility awareness has an obvious effect on such consumption behavior. Responsibility awareness is, thus, indeed an important variable affecting effective climate behavior, environmental protection, or sustainable consumption behavior.

Regarding risk awareness, Lacroix and Gifford [33] constructed a conceptual model to evaluate the frequency of low-carbon behavior and tested the correlation between cultural cognition worldview, climate change risk perception, and pro-environmental behavior barriers. The results showed that climate change risk perception is the strongest predictor of perceived barriers and energy conservation behavior frequency. Bradley et al. [34] studied the relationship between climate change risk perception and pro-environmental behavior among Australian and French residents, showing a significant correlation between the two. Lin et al. [35] discussed the difference in the impact of environmental risk perception of urban and rural residents on environmental behavior and its internal mechanism and concluded that the environmental risk perception of urban and rural residents has a significant positive impact on environmental behavior. Maartensson and Loi [36] explored the relationship between risk perception, behavioral intention, constructive hope, and pro-environmental behavior, finding that risk perception is positively correlated with behavioral intention and pro-environmental behavior. Based on a study of the path through which public environmental risk perception influences behavioral choice, Wang [37] concluded that there is a significant positive correlation between them. Hence, the public’s risk awareness is one of the important variables that determine their environmental behavior choices.

Based on previous studies, it is predicted that the four awareness dimensions—namely, climate change and energy issue perception, low-carbon awareness, personal responsibility awareness, and perception of the risks of man-made meat—have significant direct influences on the public’s intention to consume man-made meat. Consequently, the following hypotheses are proposed.

**Hypothesis** **1.**
*Perceptions of climate change and energy issues have a significant effect on the public’s intention to consume man-made meat.*


**Hypothesis 2.** 
*Low-carbon awareness will positively condition the public’s intention to consume man-made meat.*


**Hypothesis** **3.**
*Awareness of personal social responsibility will positively condition the public’s intention to consume man-made meat.*


**Hypothesis** **4.**
*Perception of the risks associated with man-made meat will positively condition the public’s intention to consume man-made meat.*


In addition to the above four dimensions, the interaction effects between different dimensions of awareness cannot be ignored. Interaction effect refers to the situation that arises when two or more factors influence the dependent variable, in which their influence is not only their influence but also the additional influence generated by their “collaboration”. For instance, the influence of the perceptions of climate change and energy issues on the intention to consume man-made meat may vary with individuals’ responsibility awareness. For this reason, the study proposes the following hypothesis:

**Hypothesis** **5.**
*Significant pairwise interactions exist between the various dimensions of awareness.*


#### 2.2.2. Moderating Effect of ID on the Awareness–Behavior Relationship

Some scholars have shown that the conditions needed for awareness to develop into behavioral intention are influenced by external situational variables that have a moderating effect (with different directions or strengths) on the awareness–behavior relationship (Wang, 2013). Given current information diversification, information influences individuals’ perception of things and the resulting behavioral intentions at all times. Human judgments are not always rational but rely on people’s automatically activated perceptions or the organization of information in their minds, external resources and opportunities, and internal awareness to determine their behavioral intentions. Numerous studies have found that access to environmental information influences people’s environmental behavior intention, which in turn affects their decisions on environmental issues. Taking Zhejiang, China, as an example, Yang [38] explored the influence of environmental ID on the pro-environmental behavior of rural residents and found a positive effect. Based on research on the role of environmental ID in 50 U.S. states, Abel [39] pointed out that if the earlier certain environmental information is disclosed to the public, the public’s understanding and participation in the environment will provide more incentives to improve environmental governance. Thus, the following is posited.

**Hypothesis** **6.**
*The disclosure of information about man-made meat will positively moderate the awareness–behavior relationship.*


To investigate the tenability of the above hypotheses, a model of how man-made meat ID influences the public’s intention to consume such products was designed (Figure 1). The influence of ID on man-made meat consumption intention is analyzed using the scale measuring consumption intention and the scales measuring the man-made meat information disclosure, perceptions of climate change and energy issues, low-carbon awareness, awareness of social responsibility, and perceived risks of man-made meat. The scales contain six subject variables and 23 observation indicators, which are divided into six groups to measure the specific meanings of the six subject variables.

## 3. Methodology

### 3.1. Data Sources

To investigate the influence of ID on the public’s intention to consume low-carbon food, a questionnaire was used to collect data with man-made meat as an example. The questionnaire was designed based on similar research articles [12,13,18]. The language used in the questionnaire is Mandarin. The questionnaire consists of two parts. The first part of the questionnaire collects socio-demographic characteristics about the respondents, such as age, gender, household size, education level, and income level; the other part covers residents’ level of knowledge about man-made meat, perceptions of climate change and energy issues, low-carbon awareness, awareness of individual social responsibility, perceived risks of man-made meat, and man-made meat consumption intention (Table 1). 

Given that the disclosure of man-made meat information cannot be directly measured, the degree of residents’ knowledge of man-made meat is used to indicate the degree of ID. Four indicators were selected—namely, the brands, ingredients, nutrition information, and price of man-made meat—to characterize the degree of information disclosure. All items were scored on a five-point Likert scale. The scores were based on individuals’ subjective assessment and represent the degree of man-made meat information disclosure (respondents’ knowledge level of man-made meat). Here, 1 means “rarely know,” 2 means “somewhat do not know,” 3 means “average,” 4 means “somewhat know,” and 5 means “know very well.” For other questions, 1 means “strongly disagree,” 2 means “somewhat disagree,” 3 means “neither agree nor disagree,” 4 means “somewhat agree,” and 5 means “strongly agree.”

Man-made meat is a new product and is still in the market exploration stage in China. Therefore, urban residents were the target population in this study. The questionnaires were distributed to urban residents in seven provincial regions: Zhejiang, Shanghai, Shandong, Anhui, Hubei, Shaanxi, and Gansu. In the midst of the COVID-19 epidemic, online communication software and a professional questionnaire survey agency, WJX.cn, were used to randomly distribute questionnaires to urban residents (the scope of the investigation was confined to provincial capital cities with man-made meat markets). The survey was conducted in June 2022, and a total of 683 valid samples were returned. Questionnaires were considered invalid if respondents did not complete the survey or if their response time exceeded the median of those of all respondents. This method was used to remove invalid questionnaires, and 647 valid questionnaires were finally obtained, with a valid return rate of 94.73%.

Of the 647 respondents, 384 were male (59.35%), and 263 were female (40.65%). Ages ranged from 17 to 60 years old, with an average age of 27 years old, which was basically in line with the demographic characteristics of the Chinese urban population with spending power. In addition, the sample included public institution personnel (17%), corporate employees (20%), college students (32%), administrative personnel (13%), retirees (10.8%), freelancers (5.2%), and others (2%), ensuring the sociodemographic diversity of respondents.

### 3.2. Data Tests

To ensure data quality, the observable variables were tested using SPSS Statistics 21 software (IBM Corp., Armonk, NY, USA). The reliability coefficient of each latent variable was between 0.617 and 0.935, larger than the suggested threshold value of 0.6.

Before formally forming the scale, experts in psychology and behavioral economics and representative members of the public were invited to conduct in-depth interviews on factors important for explanatory and outcome variables, based on which the original scale was obtained. A pre-survey was then administered to the public, followed by an analysis of the pre-survey results and a summary of the constructive comments of the respondents, further improving the scale.

This study used factor analysis to test the construct validity. The value of the validity test ranged from 0.547 and 0.834, larger than the suggested threshold value of 0.5 (with the Bartlett sphericity test less than 0.00). The results of the reliability and validity tests indicated that the sample data were of good quality and passed the tests.

## 4. Results

### 4.1. Model Validation

The interdependence of the variables is represented by the Pearson correlation coefficient matrix (Table 2).

In terms of awareness, the scores of the dimensions of SRA, CCEIP, and LA are relatively high (with mean values greater than 3.7), and the score for MMCI is relatively low. Hence, the public pays more attention to resource and environmental issues. At the same time, the public has a strong sense of responsibility for the environment but does not have a high MMCI. The public has a low level of knowledge about man-made meat, with a score of only 2.417. Therefore, the current level of ID related to man-made meat is low, and the public does not know much about the sources, brands, composition, nutritional value, and price of man-made meat.

LA, SRA, and MMRP were significantly correlated with the public’s MMCI at the 0.001 level. CCEIP and the public’s MMCI were significantly correlated at the 0.01 level. In terms of the Pearson correlation coefficients between the various variables and behavior, MMRP is the largest, followed by LA and SRA, while CCEIP is the smallest.

A causal model of the effect of MMID on the public’s MMCI was constructed using structural equations. The influence of ID on the public’s MMCI was analyzed by taking CCEIP, LA, SRA, and MMRP as independent variables and MMID as moderating variable. In Hypotheses 1, 2, 3, and 4, this study constructs three possible influence paths of “awareness → behavioral intention”, namely “CCEIP → MMCI”, “LA → MMCI”, “SRA → MMCI”, and “MMRP → MMCI”. In Hypothesis 5, four paths of interaction between the dimensions of awareness are designed. In Hypothesis 6, four moderating paths of “awareness → situation → behavioral intention” were constructed; that is, the path through which MMID moderates “awareness → behavioral intention”.

The model showed good performance in the initial test; however, it was found from the significance test of the model that the path of MMID → SRA and the path of CCEIP → MMCI were less significant. Hence, the optimal model was obtained by continuous model revision (Figure 2).

Further tests of the structural equation model found that the chi-square degree of freedom ratio of the model is 2.921 < 3.00 (*p* = 0.000), and the CN value is 296 > 200; other tests were performed, as shown in Table 3. Therefore, the hypothesis model has a good overall fit and passes the robustness test.

### 4.2. Main Effects of Awareness on Behavioral Intention and Interaction Effects for Awareness

The model simulation results show that among the four dimensions of awareness, SRA, LA, and MMRP all influence the public’s MMCI, except for CCEIP (Table 4). Risk perception has the greatest influence. At the same time, there is a negative correlation between MMRP and MMCI (*p* < 0.001), indicating that the higher the public’s perception of the risks of man-made meat, the less inclined they are to consume such products. LA is positively correlated with the public’s intention to consume man-made meat (*p* < 0.01), indicating that individuals with stronger LA are more inclined to change their dietary habits and eventually achieve carbon reduction. SRA is positively correlated with the public’s intention to consume man-made meat at the 0.01 level, indicating that individuals who are more willing to take responsibility for resources and environmental protection tend to try to consume man-made meat. Theoretical hypotheses 2, 3, and 4 are thus confirmed, while theoretical hypothesis 1 does not hold.

The results of the interaction effects among the three dimensions of awareness—namely, LA, SRA, and MMRP—show that only the interaction item of LA and MMRP has a significant interaction effect on the public’s MMCI (hence hypothesis 5 partially holds). The interaction between LA and MMRP is shown in Figure 3. In Figure 3a, LA is divided into high LA and low LA to analyze the relationship between residents’ LA and MMCI. The study finds that MMRP has a stronger negative effect on the MMCI for the public with higher LA, so enhancing their LA can more effectively promote their MMCI. In comparison, MMRP has a weaker negative effect on the MMCI for individuals with lower LA. In Figure 3b, the MMRP is divided into high and low values to analyze the relationship between LA and the public’s MMCI. It is found that LA has a weaker positive effect on the MMCI for the public with a higher MMRP. Reducing the public’s risk perception can thus more effectively promote their MMCI. In comparison, LA has a stronger positive effect on the MMCI for individuals with lower MMRP. The policy implication of this is that when developing measures to enhance the public’s low-carbon consumption intention, policy makers should not only improve the public’s personal LA but also reduce their risk perception, which would improve the utility of the policy.

### 4.3. Moderating Effect

According to the path analysis results of the structural equation model (Figure 2), it can be observed that ID has the most significant moderating effect on the relationship between LA and the public’s MMCI, followed by that on the relationship between MMRP and the public’s MMCI, but ID has no significant moderating effect on the relationship between SRA and the public’s MMCI (hypothesis 6 partially holds). Therefore, MMID promotes the public’s MMCI by raising their LA. In addition, ID diminishes the public’s perception of the risks of man-made meat, thereby promoting their consumption intention.

The interaction between MMID and LA is shown in Figure 4a. MMID is divided into high and low groups to analyze the moderating effect of MMID on the relationship between LA and the public’s MMCI. It is found that the positive relationship between LA and the MMCI is stronger for respondents with more knowledge about man-made meat and weaker for those with less knowledge. Therefore, enhancing the degree of ID can more effectively promote the public’s MMCI.

The interaction between MMID and MMRP is shown in Figure 4b. MMID is divided into high and low groups to analyze the moderating effect of MMID on the relationship between MMRP and the public’s MMCI. It is found that the negative relationship between MMRP and the MMCI is weaker for respondents with more knowledge of man-made meat and stronger for those with less knowledge. That is to say; for the public with a deeper knowledge of man-made meat, their risk perception has less influence on their intention to consume such meat. Enhancing ID can thus effectively reduce the influence of risk perception on the intention to consume man-made meat and promote the public’s consumption intention.

To more clearly identify the moderating effect of MMID on the public’s MMCI, the moderating effect of each factor of MMID on consumption intention is analyzed. The results indicate that ID influences the public’s MMCI by affecting LA, while nutrition information is the most important factor influencing the relationship between MMRP and consumption intention. Therefore, the moderating effect of MMID on the MMCI varies greatly with the awareness dimension.

## 5. Discussion

### 5.1. Direct Influence of ID on Consumption Intention

The results show a direct relationship between MMID and the public’s MMCI, which is consistent with the findings of most scholars [40]. However, this study only examined the four aspects of “brand, ingredients, price, and nutrition”, and did not differentiate the differences in the impact of different types of information. Wang et al. [41] studied what information is critical to Beijing consumers’ intention to purchase man-made meat and found that purchase intention increases significantly after nutritional information is provided. Van Loo et al. [42] showed that, compared with the case of traditional animal meat, environmental information has little influence on the original market share but promotes more new consumers to enter the market.

In addition, the frequency and the way the information is presented also influence consumers’ purchase intentions. Wichman [43] studied the relationship between information provision and consumer behavior and found that frequent information provision stimulates public consumption. Bekker et al. [44] found that positive and negative descriptive information leads to differences in consumers’ direct attitudes toward man-made meat. Siegrist and Sütterlin [45] reported that the descriptions of man-made meat without technical terms increase consumer acceptance, while descriptions with technical terms cause consumers to perceive unnaturalness and lead to aversion. Unlike ID for general consumption behavior, the disclosure of corresponding information about man-made meat as a low-carbon consumer product can increase the public’s sense of responsibility and urgency, thereby promoting their intention to engage in pro-environmental behavior.

### 5.2. Direct Influence of Awareness on Consumption Intention

LA and SRA are the basis for the generation of behavioral MMCI by influencing individuals’ psychological preference for low-carbon consumption. When individuals lack awareness, they inevitably do not consciously generate any behavioral intention for low-carbon consumption. Ajzen and Madden [17] proposed that certain behavioral intention of the public is affected by their environmental perception. Ye and Mattila [46] investigated the influence of environmental information on consumers’ responses to man-made meat and found that the perceived association between meat consumption and climate change influences consumer attitudes toward man-made meat products and their purchase intention. Cliceri et al. [47] compared the attitudes of people with different dietary preferences toward man-made and animal-based dishes and found that food awareness plays an important role in determining dietary habits. However, some studies have reached opposite conclusions. Kopplin and Rausch [48] examined the role of consumer dietary behavior in purchasing man-made food substitutes and did not find that environmental concerns, consumers’ perceived effectiveness, and health awareness influence dietary behaviors.

Although LA and SRA, which are internal dimensions of the awareness structure, are correlated with man-made meat consumption behavior, our study did not find a significant positive interaction between the two. Some studies have shown that different types of awareness are not independent of each other but interact with each other. Based on a study of the pro-environmental behavior of rural residents under comprehensive environmental management, Wang [12] found that groups with a high sense of personal responsibility for environmental issues are more likely to increase their preference for pro-environmental behavior choices in the public domain if their positive environmental attitudes are cultivated. Hou et al. [18] investigated the public’s intention to reuse reclaimed water and found that the dimensions of water conservation awareness and personal responsibility awareness exist in a significant positive interaction—that is, water conservation awareness plays a role in amplifying awareness of personal responsibility. There are two possible reasons for the inconsistent findings in this study. First, energy conservation and carbon reduction are sustainable development strategies advocated worldwide in recent years, and the public is not clear about the relationship between their daily life behavior and carbon reduction. Second, there is a perception among some members of the public that “enterprises are primarily responsible for energy conservation and carbon reduction efforts, as they are the main contributors to carbon emissions”. This belief suggests that residents have a minimal role in these efforts (38.7%). Strengthening the public’s awareness of personal responsibility for carbon reduction should thus be the direction of future policy efforts.

No consensus has been reached in academia on the influence of resource issue awareness on resource behavior. Hou et al. [18] found that there is no direct relationship between the perception of water resources and environmental issues and the intention to reuse reclaimed water, while Wang et al. [13] found that the awareness of resource issues makes a positive contribution to resource behavior. The results of the present study show that there is no direct relationship between CCEIP and MMCI. The reason may be that the majority of the public believes that climate change and energy-related issues are responsibilities that should be taken by governments at a national level and are beyond the limited behavioral power of a person, or there are doubts about the carbon reduction effect of man-made meat. In addition, people’s perceptions and attitudes toward climate change and energy issues have become relatively clear, and most people agree that the situation of environmental issues is very serious. However, the climate change and energy issues designed in the present study are relatively broad and not specific to particular environmental issues, so it is difficult for most people’s perceptions of the general environment to correspond directly to a specific environmental protection behavior.

### 5.3. Moderating Effect of ID on Awareness and Consumption

Economists have argued that ID plays an important role in people’s daily behavioral decision-making by enlightening our behavior planning. However, in real life, information asymmetry is very common. The results of this study show that ID related to man-made meat had a significant moderating and magnifying effect on the relationship between LA and the MMCI. In addition, it was noted in other studies that ID not only has an important influence on the formation and improvement of public environmental awareness but also leads to “adverse selection” in consumption and investment by the public (Huang, 2011). For example, in the case of asymmetric information, high-quality products will have a greater positive externality compared with low-quality products, enabling low-quality products to exist in the market and obtain prices higher than their own.

This study found that ID had a significant moderating effect on the relationship between the public’s perception of the risks of man-made meat and their intention to consume, as well as a weakening effect on the relationship between risk perception and intention to consume. In general, individuals’ risk awareness and risk perception were not entirely acquired through direct, first-hand experience. When the risks of things are unknown, individuals exaggerate their perception of those risks for their own safety. In the fast-developing information modern society, new information communication media (represented by the Internet) can reduce the public’s perception of relevant risk in the process of reporting and explaining risk events. Other studies have also found that the more informed the public is about man-made meat, the more likely their rejection of man-made meat will be alleviated and the greater their intention to consume man-made meat [49]. Slade [50] concluded that the perception of health risks negatively influences behavioral intention, so increased knowledge of man-made meat information weakens the perception of health risks to a certain extent, thus promoting consumer acceptance of man-made meat.

## 6. Conclusions and Recommendations

### 6.1. Conclusions

This study analyzed the path of the effect of ID on the public’s MMCI by constructing a structural equation model of the influence of ID on MMCI. The following conclusions are drawn:(1)LA, SRA, and MMRP significantly influence the public’s MMCI, with MMRP having the greatest influence.(2)LA and MMRP have a significant interaction effect on the public’s MMCI.(3)ID has the most significant moderating effect on the relationship between LA and the MMCI, followed by its moderating effect on the relationship between MMRP and MMCI.

### 6.2. Policy Recommendations

Based on the above findings, the following recommendations are proposed for how the government can promote the public’s MMCI by increasing the degree of ID and the public’s LA, as well as reducing the perception of the risks associated with man-made meat. First, the public’s psychological perception of climate change and environmental crisis issues should be effectively influenced by strengthening the publicity around climate change and environmental issues in various ways (e.g., theme-based education, knowledge contests, visits, and community consultations). By communicating the seriousness and reality of the climate change situation and the rapid increase in personal harm, the public will be impressed and motivated to act quickly to change their consumption patterns. Second, the public’s awareness of responsibility for social and environmental crisis issues should be cultivated and increased. In the face of social and environmental crises, the tendency to shift or diffuse responsibility is widespread. Policy makers should thus clarify the rights and responsibilities of individuals related to environmental crisis issues to reduce contextual ambiguity while strengthening formal and informal supervision to increase individual responsibility. The public’s knowledge of low-carbon consumption and behavior guidelines should also be more broadly popularized. According to our in-depth interviews, many people had a rather vague concept of low-carbon consumption and were unclear about the logic of man-made meat and carbon emissions reduction. Popularizing knowledge of low-carbon consumption and informing the public of the operational guidelines for low-carbon consumption patterns are essential to increase awareness. Finally, the public’s perception of the effects of their individual behavior should be improved. Policy makers should try to avoid making the public feel that there is imminent total collapse or that they are powerless and unable to turn back the clock on environmental crisis issues such as global climate change. Instead, policy makers should focus on emphasizing the significant positive effects of changing the public’s consumption patterns and lifestyle.

## Figures and Tables

**Figure 1 ijerph-20-02950-f001:**
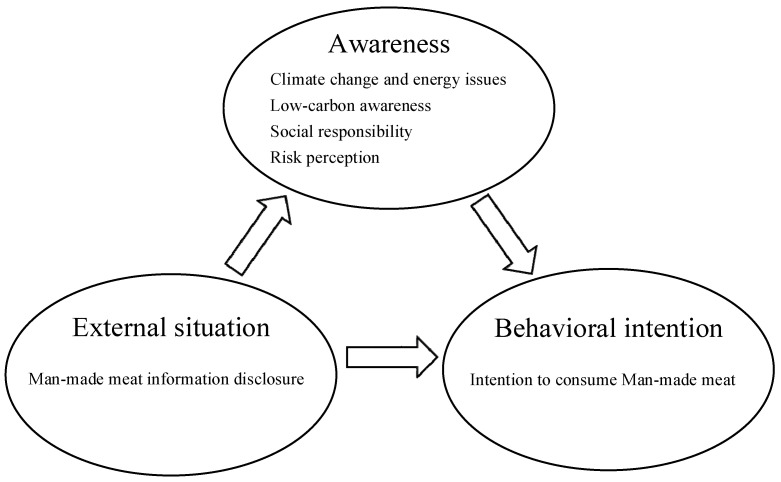
The awareness-situation-behavior model.

**Figure 2 ijerph-20-02950-f002:**
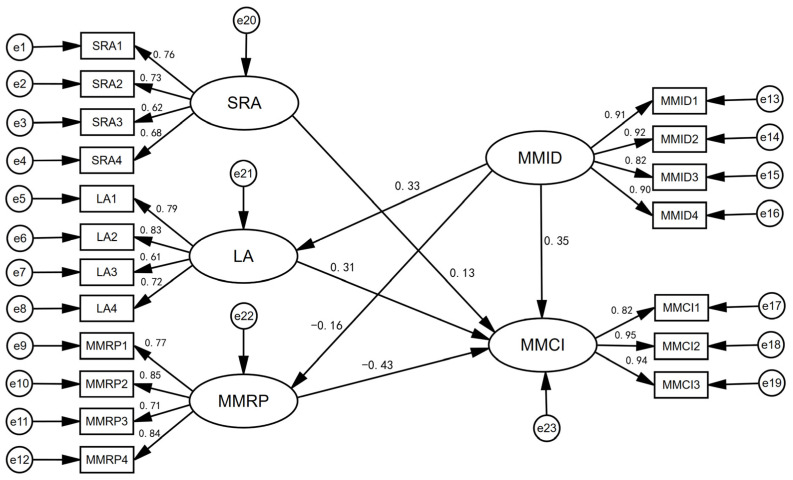
The structural equation model of the public’s MMCI.

**Figure 3 ijerph-20-02950-f003:**
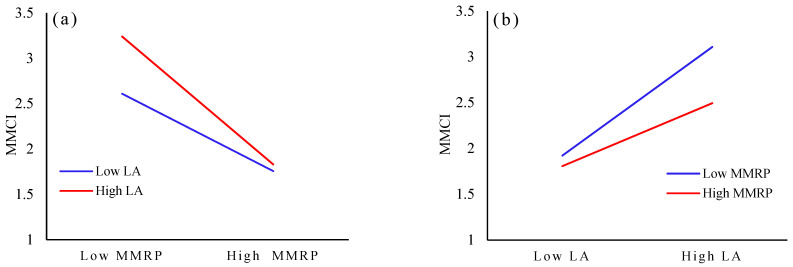
The Interaction between LA and MMRP. (**a**) The effect of MMRP on LA; (**b**) The effect of LA on MMRP.

**Figure 4 ijerph-20-02950-f004:**
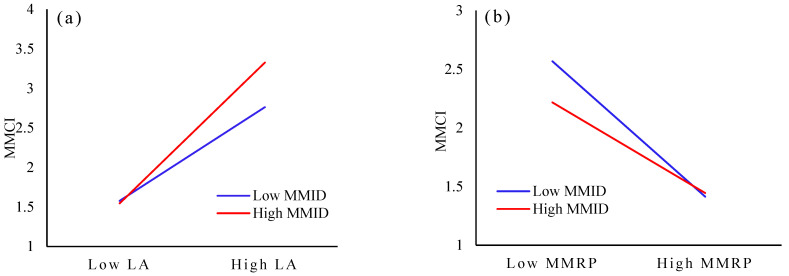
The interaction between MMID and LA and MMRP. (**a**) The effect of LA on MMID; (**b**) The effect of MMRP on MMID.

**Table 1 ijerph-20-02950-t001:** The indicator system of factors influencing the public’s intention to accept man-made meat.

Level	Dimension	Indicator	Code	Mean Value	Standard Deviation	Reliability	KMO
Awareness level	Climate change and energy issue perception (CCEIP)	The current energy situation in China is very tight	CCEIP1	4.300	0.837	0.617	0.547
		We will soon face an ecological disaster if global warming continues	CCEIP2	3.867	1.167		
		Advocating low-carbon behavior can help combat climate change	CCEIP3	4.467	0.776		
		There is enough clean energy on Earth to replace traditional energy as long as we learn how to develop clean energy	CCEIP4	2.933	1.507		
	Low-carbon awareness (LA)	I am conscious of saving electricity	LA1	4.233	0.858	0.813	0.796
		I usually choose public transportation unless I have to use other modes	LA2	3.733	0.874		
		I would buy used products such as books and electronics	LA3	3.433	0.971		
		I am willing to spend a little more money on low-carbon products	LA4	3.433	0.925		
	Social responsibility awareness (SRA)	I feel obligated to contribute to the “dual carbon” goal	SRA1	4.300	0.702	0.747	0.770
		Low-carbon life is the responsibility of every citizen	SRA2	4.300	0.750		
		Enterprises should take the initiative to take responsibility for energy conservation and carbon reduction	SRA3	3.500	0.732		
		Low-carbon consumption is a manifestation of social responsibility	SRA4	4.367	0.669		
	Man-made meat risk perception (MMRP)	Consumption of man-made meat may pose unknown risks such as obesity	MMRP1	3.300	0.837	0.867	0.800
		Eating man-made meat can affect my health	MMRP2	3.200	0.714		
		The thought of eating man-made meat makes me feel sick	MMRP3	3.100	0.803		
		Man-made meat is junk food	MMRP4	2.900	0.803		
Behavioral intention level	Intention to consume man-made meat (MMCI)	I am willing to try eating man-made meat	MMCI1	2.667	0.994	0.921	0.729
		I would actively consume man-made meat	MMCI2	2.233	0.916		
		Consuming man-made meat gives me a sense of honor	MMCI3	2.333	0.941		
External contextual level	Man-made meat information disclosure (MMID)	Man-made meat brands such as Starfield and Beyond Meat	MMID1	2.400	0.937	0.935	0.834
		Ingredients of man-made meat	MMID2	2.567	1.019		
		Nutrition of man-made meat	MMID3	2.300	0.988		
		Price of man-made meat	MMID4	2.400	1.016		

**Table 2 ijerph-20-02950-t002:** Correlation coefficient matrix between variables.

	CCEIP	LA	SRA	MMRP	MMCI	MMID
Mean (M)	3.892	3.708	4.117	3.125	2.411	2.417
S.D.	0.567	0.910	0.675	0.669	0.793	0.923
CCEIP	1					
LA	0.141 **	1				
SRA	0.074	0.503 ***	1			
MMRP	0.014	−0.289 **	−0.172 **	1		
MMCI	0.164 **	0.486 ***	0.387 ***	−0.506 ***	1	
MMID	0.299 **	0.302 **	0.198	−0.185 **	0.498 ***	1

Note: *** shows mean difference is significant at the 0.001 level. ** shows mean difference is significant at the 0.01 level.

**Table 3 ijerph-20-02950-t003:** Results of model fitting test.

Type of Index	Statistics	Standard Value	Test Value	Adaptability of the Model
Absolute goodness-of-fit	χ^2^/df	<3.00	2.921	Qualified
	χ^2^	*p* < 0.05	*p* = 0.000	Qualified
	RMSEA	<0.05	0.046	Qualified
Value added goodness-of-fit	CFI	>0.90	0.964	Qualified
	NFI	>0.90	0.926	Qualified
	IFI	>0.90	0.965	Qualified
	RFI	>0.90	0.904	Qualified
Concise goodness-of-fit	PNFI	>0.50	0.748	Qualified
	PCFI	>0.50	0.780	Qualified
	CN	>200	296	Qualified

**Table 4 ijerph-20-02950-t004:** Interaction between awareness.

The Main Effect	MMCI	The Interaction Effect	MMCI
SRA	0.127 **	LA×SRA	0.033
LA	0.307 **	MMRP×SRA	−0.465
MMRP	−0.434 ***	MMRP×LA	−0.694 **
MMID	0.347 ***	—	—

Note: *** *p* < 0.001, ** *p* < 0.01.

## Data Availability

The research data can be obtained from the first author after the funding project is completed.

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
