# Peer review of "Information Disclosure Impacts Intention to Consume Man-Made Meat: Evidence from Urban Residents’ Intention to Man-Made Meat in China"

_ijerph, 2023, doi:10.3390/ijerph20042950_

Round 1

Reviewer 1 Report

I have read the paper with interest. While the results are of potential interest, I don't think this paper should be published as the theoretical model is not based on economic theory, and the results are based on a hypothetical, unincentivized survey. I wouldn't have enough trust in these data to recommend moving forward with the manuscript. My best advice is that the authors could send the paper to a journal somewhat outside of Economics, which might be more receptive to this type of work. In addition, the authors should identify their contribution and distinguish it with previous studies. The authors mention few studies explored the relationship between public’s intention from an integrated perspective and the pathways through which they act on consumption intention. I feel this statement is more like a description of the paper rather than a contribution. 

Reviewer 2 Report

§  The manuscript entitled ‘Impacts of Information Disclosure on the Public's Intention to Consume Man-made Food in the Context of Health Economics: Evidence from Urban Residents’ Intention to Accept Man-made Meat in China’ presents an interesting and trending issue, however, some corrections are needed.

§  The title can be improved and shorter

§  In the abstract, the methodology used in this study has to be included.

§  It is better not to use “Because” … at the beginning of sentences.

§  “647 respondents” - How was the sample size established? The sample size is slightly small compared to the population of seven provincial regions in China.

§  What language does the author use in the questionnaire? (English, Mandarin, or other)

§  Was the questionnaire adapted from a previously published study? What was the language used in the questionnaire? Was the questionnaire translated?

§  L 47-52. Adding reference is needed.

§  L 61. Adding a reference is needed

§  L 321. It is better to use the term “socio-demographic characteristics” instead of basic information.

§  L334. “For other items” this sentence is not clear

§  L 374. The title of table 2. need more information

Reviewer 3 Report

The manuscript describes research conducted to explore the effect of ‘information disclosure’ on the intention to the consumption of ‘man-made food’. The objective of this study could be much clear and therefore will need some review.  The title will require reviewing to reflect the content of the manuscript. It is not clear what ‘information disclosure’ in the title actually mean. Is it supposed to mean disclosure of information from the manufacturer of the man-made meat? Or on the part of the consumer? In my view, ‘information disclosure’ should be replaced by ‘consumer awareness’ which is more appropriate considering the general overview of information provided in the manuscript. I will also suggest authors replace ‘man-made food’, in the title, with ‘man-made meat’, however, authors will also have to take a second look at the use of the phrase ‘man-made meat’. Wouldn’t ‘cultured meat’ or ‘lab-grown meat’ be more appropriate in line with majority publication in the literature?

Abstract: will need reviewing, particularly, the methodology of the study as well as summary of data and the main findings of the study. It will be useful to present some summary of the data from the results.

Introduction: Introduction should be shortened for a better perception of the information by readers. The entire section will need re-written to shorten it and also for clarity. It is not clear why literature review is included. This information could be better used in the discussion and some integrated into the shortened introduction, particularly the hypothesis (must also be in line with the aim of the study). There are some sentences that are incomplete and authors much check and rectify this. E.g. lines 47-48. Some sections will require references: e.g. lines 58-61, 74-78.

Methodology: The methodology must be written in past tense. Experimental design can be confusing at times, and it is because it is mixed with results in some instances. The section will need restructuring for continuity and clarity. Is the table 1 for methodology or about results or? It will be useful to define the codes used in the tables in the methodology before they are applied in the tables. The definition of the codes is presented in lines 396-409, which is further away from when they were first used and in the results. Is information provided in lines348-354 results or methodology?

Line 321: check ‘one part collections basic’.

Results and Discussion: If possible, I will suggest authors to combine the results with discussion section. As it stands, most part of the discussion is just presenting results of other published studies and not discussion of the results for the current study. Especially, lines 507-571, 595-609.

Will also suggest authors to use third person tenses instead of first person.

Line 446: it is not clear what authors mean by ‘weaker negative’.

Reviewer 4 Report

Information about the nutritional values of Man-made meat and normal meat can be given. Bekker et al (2017) are cited twice in the article. These resources should be given as 2017a and 2107b in both in the article and in the references.

Round 2

Reviewer 1 Report

The paper can be accepted for publication provided it will be scientifically edited to follow the strict instructions of the journal.

Author Response

Thanks for your comments. The paper will be scientifically edited to follow the strict instructions of the journal.

Reviewer 3 Report

Authors have made a good effort to revise the manuscript. However, there is the need to further revision before it can be published.

The use of the phrase 'Information disclosure' in the title still makes it unclear. It sounds like the information is being disclosed to consumers by another party and not necessarily consumers developing their knowledge using information available to them, which will create the awareness and therefore determine their intensions, in this case, about man-made meat. Information disclosure is vague in this case, in my opinion.

I strongly recommend the authors to reconsider the literature review content. This hypothesis could be included in the introduction and theories could be used to support the design of the research in the methodology. 

The codes used in the tables in the methodology must be define before it was used. The definitions, as indicated by the authors, come after  those codes have been used which is not appropriate (in the results section).

Authors to still check and ensure grammar in line 332 is amended: 'One part collections socio-demographic characteristics......'
